# In Vitro Combinations of Baloxavir Acid and Other Inhibitors against Seasonal Influenza A Viruses

**DOI:** 10.3390/v12101139

**Published:** 2020-10-08

**Authors:** Liva Checkmahomed, Blandine Padey, Andrés Pizzorno, Olivier Terrier, Manuel Rosa-Calatrava, Yacine Abed, Mariana Baz, Guy Boivin

**Affiliations:** 1Centre Hospitalier Universitaire de Québec—Centre Hospitalier de l’Université Laval (CHUQ-CHUL) and Laval University, Québec City, QC G1V 4G2, Canada; liva.checkmahomed@crchudequebec.ulaval.ca (L.C.); yacine.abed@crchudequebec.ulaval.ca (Y.A.); 2CIRI—Centre International de Recherche en Infectiologie, (Team VirPath), University Lyon, Inserm U1111, Université Claude Bernard Lyon 1, CNRS—Centre National de la Recherche Scientifique UMR5308, ENS de Lyon, F-69007 Lyon, France; blandine.padey@univ-lyon1.fr (B.P.); mario-andres.pizzorno@univ-lyon1.fr (A.P.); olivier.terrier@univ-lyon1.fr (O.T.); manuel.rosa-calatrava@univ-lyon1.fr (M.R.-C.); 3Signia Therapeutics SAS, 69100 Villeurbane, France; 4VirNext, Faculté de Médecine RTH Laennec, Université Claude Bernard Lyon 1, Université de Lyon, 69008 Lyon, France

**Keywords:** baloxavir, combination, influenza, A(H1N1) virus, A(H3N2) virus, polymerase inhibitors, neuraminidase inhibitors, human airway epithelium

## Abstract

Two antiviral classes, the neuraminidase inhibitors (NAIs) and polymerase inhibitors (baloxavir marboxil and favipiravir) can be used to prevent and treat influenza infections during seasonal epidemics and pandemics. However, prolonged treatment may lead to the emergence of drug resistance. Therapeutic combinations constitute an alternative to prevent resistance and reduce antiviral doses. Therefore, we evaluated in vitro combinations of baloxavir acid (BXA) and other approved drugs against influenza A(H1N1)pdm09 and A(H3N2) subtypes. The determination of an effective concentration inhibiting virus cytopathic effects by 50% (EC50) for each drug and combination indexes (CIs) were based on cell viability. CompuSyn software was used to determine synergism, additivity or antagonism between drugs. Combinations of BXA and NAIs or favipiravir had synergistic effects on cell viability against the two influenza A subtypes. Those effects were confirmed using a physiological and predictive ex vivo reconstructed human airway epithelium model. On the other hand, the combination of BXA and ribavirin showed mixed results. Overall, BXA stands as a good candidate for combination with several existing drugs, notably oseltamivir and favipiravir, to improve in vitro antiviral activity. These results should be considered for further animal and clinical evaluations.

## 1. Introduction

Since 1977, two subtypes of influenza A viruses, H1N1 and H3N2, along with influenza B viruses, have co-circulated in the human population. These viruses are responsible for seasonal epidemics and may cause global pandemics [1]. Influenza epidemics can lead to millions of cases of severe respiratory illnesses and to half a million deaths annually worldwide [2] representing a major public health challenge. Vaccination still remains the first line of prevention against seasonal influenza infections [3,4,5] but the protection conferred by actual vaccines varies from year to year due to the evolution of the circulating strain and the vaccine match [6,7]. Antivirals are recommended for the treatment of seasonal influenza viruses, especially in high risk populations (elderly, immunocompromised patients and subjects with co-morbidities) [8]. Neuraminidase inhibitors (NAIs), such as oseltamivir, zanamivir, peramivir and lanamivir, constitute the main recommended drugs for the treatment of influenza viruses in many countries [9,10,11]. These drugs inhibit the neuraminidase (a glycoprotein that enables the virus to be released from host cell) which reduces clinical illness [9,12]. In addition to this major class of antivirals, RNA-dependent RNA polymerase (RdRp) inhibitors are also approved in a few countries to treat influenza A and B viruses. Favipiravir (also known as T-705) is a purine nucleoside analog that is recognized as an alternative substrate by the viral polymerase and is incorporated into the nascent RNA, leading to RNA viral replication errors [13,14]. Recently, baloxavir marboxil (BXM; prodrug of baloxavir acid, BXA), a cap-dependent endonuclease inhibitor, has been licensed in a few countries including USA. This compound targets the endonuclease activity of the polymerase acidic (PA) subunit, resulting in the inhibition of RNA transcription and replication [14]. Not surprisingly, prolonged antiviral therapy may lead to the emergence of drug resistance. Indeed, H275Y, E119G, I427T or I223R (N1 numbering) substitutions, among others in the neuraminidase (NA) protein, induce single or multi-drug resistance [15,16,17]. Rapidly after licensure, the I38T PA substitution was shown to reduce sensitivity to BXM [18,19,20,21]. On the other hand, the K229R substitution in motif F of the PB1 subunit confers in vitro resistance to favipiravir [22]. Several in vitro and in vivo studies have shown that dual antiviral therapies may have a significant positive effect on the treatment of influenza infections [13,23,24,25] and could further delay the emergence of resistance. Indeed, combination therapy with BXM and oseltamivir was reported to produce synergistic responses against the influenza A/PR/8/34 strain and to reduce virus titers [26]. However, there is a lack of information about combination therapy between this new polymerase inhibitor and other approved drugs. Moreover, the impact of combining two polymerase inhibitors is still unknown. Therefore, we assessed the inhibitory effects of BXA with three NAIs (oseltamivir, zanamivir and peramivir) or two other polymerase inhibitors (favipiravir and ribavirin) in cell cultures and in human airway epithelia (HAE) infected with influenza A(H1N1)pdm09 and A(H3N2) viruses. 

## 2. Materials and Methods

### 2.1. Cells, Viruses and Compounds

Madin–Darby canine kidney cells over expressing the α2,6 sialic acid receptor (ST6-GalI-MDCK cells) (kindly provided by Y. Kawaoka from the University of Wisconsin, Madison, WI, USA [27]) were maintained in Minimum Essential medium (MEM) supplemented with 10% fetal bovine serum (FBS), HEPES (Invitrogen, Carlsbad, CA, USA) and 7.5 µg/mL of puromycin.

MucilAirTM reconstituted HAE, issued from primary cells obtained from pools of nasal biopsies, were provided by Epithelix SARL (Geneva, Switzerland) and maintained in air–liquid interphase with specific culture medium in Costar Transwell inserts (Corning, NY, USA), according to the manufacturer’s instructions. 

The A/California/7/2009 (H1N1)pdm09, A/Switzerland/9715293/2013 (H3N2) and A/Texas/50/2012 (H3N2) influenza viruses were obtained from NIBSC (code numbers 15/252 and 14/224, respectively) and from the Centre National de Référence des Virus des Infections Respiratoires, Lyon, France Sud.

Baloxavir acid was synthesized at Shionogi & Co., Ltd. (Osaka, Japan) and diluted in dimethyl sulfoxide (DMSO) to a stock concentration of 10 mM. Favipiravir (T-705) was purchased from Adooq Bioscience (Irvine, CA, USA) and diluted in DMSO to a stock concentration of 100 mM. Oseltamivir carboxylate (the active form of oseltamivir) was synthesized by Hoffmann-La Roche (Basel, Switzerland) and diluted in sterile water to a stock concentration of 10 mM. Zanamivir and ribavirin were purchased from Sigma (St-Louis, MO, USA) and diluted in sterile water and phosphate-buffered saline (PBS), respectively, to stock concentrations of 10 mM and 100 mM, respectively. Peramivir was synthesized by Biocryst (Durham, NC, USA) and diluted in sterile water to a stock concentration of 10 mM.

### 2.2. In Vitro Studies of Antiviral Combinations

ST6-GalI-MDCK cells (5000 cells/well) were seeded in 96-well plates and inoculated with 200 Median Tissue Culture Infectious Dose per well (TCID50/well) of each of the influenza strains (A/California/7/2009 (H1N1)pdm09 and A/Switzerland/9715293/2013 (H3N2)). Infected cells were incubated for 60 min at 37 °C in a 5% CO_2_ atmosphere, followed by the addition of BXA, each NAI, favipiravir and ribavirin in two-fold serial dilutions. For A(H1N1)pdm09 infections, concentrations of 0.0175–10 nmol/L were used for BXA; 2.34–1000 nmol/L for oseltamivir; 2.34–1000 nmol/L for zanamivir; 0.468–400 nmol/L for peramivir; 78.125–80,000 nmol/L for favipiravir; 78.125–40,000 nmol/L for ribavirin, diluted in MEM and TPCK (L-1-Tosylamide-2-phenylethyl chloromethyl ketone) trypsin (1 mg/mL). For A(H3N2) infections, concentrations of 0.39–200 nmol/L were used for BXA; 7.8–4000 nmol/L for oseltamivir; 35.15–16,000 nmol/L for zanamivir; 1.56–800 nmol/L for peramivir; 312.5–160,000 nmol/L for favipiravir; 78.125–40,000 nmol/L for ribavirin, diluted in MEM and TPCK trypsin (1 mg/mL). These dilutions were based on Fukao et al. [26] and on our preliminary experiments. Forty-eight hours later, cell viability was assessed by adding 10 µL/well of 3-(4,5-dimethylthiazol-2yl)-5-(3-carboxymethoxyphenyl)-2-(4-sulfophenyl)-2H-tetrazolium, inner salt (MTS) (Cell Titer 96 Aqueous One Solution Cell Proliferation Assay, Promega, Madison, WI, USA) and absorbance was measured at 490 nm with a plate reader. The percentage of cell survival was calculated as described elsewhere [28,29]. Then, 50% effective concentration (EC50) values were calculated for each viral strain using Prism software (GraphPad, v7). For two-drug combination studies, the EC50 value of each drug was used to determine an equipotent ratio between the two compounds. The synergism, additivity or antagonism were calculated using the combination index (CI) values, as reported by Chou [30].

Briefly, the weighted average CI (CIwt) was calculated for each combination as (CI50 + 2 × CI75 + 3 × CI90 + 4 × CI95)/10 to estimate drug combination effects at high levels of virus inhibition and to increase therapeutic relevance [31]. 

Drug combination effects were defined as CIwt < 0.7, synergism; CIwt > 0.7 and <0.9, moderate synergism; CIwt > 0.9 and <1.2, additivity; CIwt > 1.2 and <1.45, moderate antagonism; CIwt > 1.45, antagonism.

### 2.3. Evaluation of Antiviral Combinations in HAE

The apical poles of HAE were gently washed twice with warm OptiMEM medium (Gibco, ThermoFisher Scientific, Gaithersburg, MD, USA)) and then infected with a 150 μL dilution of A/California/7/2009 (H1N1)pdm09 or A/Texas/50/2012 (H3N2) in OptiMEM, at a multiplicity of infection (MOI) of 0.1 or 0.01, respectively. For mock infection, the same procedure was followed using OptiMEM as inoculum. After 1 h incubation at 37 °C, the viral inoculum was removed. At 5, 24 and 48 hpi, treatments with specific dilutions of antiviral molecules alone or in combination in MucilAir^®^ culture medium (Epithelix SAS, Saint Julien-en-Genevois, France) were applied through basolateral poles. At 24, 48 (before treatment renewal) and 72 hpi, apical washes with 150 μL OptiMEM were performed and stored in two aliquots at −80 °C for TCID50 viral titration. At 24, 48 and 72 hpi, transepithelial electrical resistance (TEER) between the apical and basal poles of the HAE, considered as a surrogate of epithelium integrity, was measured using a dedicated volt-ohm meter (EVOM2) and compared to baselines values measured before infection (*t* = 0).

### 2.4. Statistical Analysis

All experimental assays were performed in triplicate at a minimum, and representative results are shown. HAE viral titers in the drug combination groups were compared against those of both the untreated (*) and most performant single-treated (#) groups using mixed model two-way analysis of variance (ANOVA) with Bonferroni post hoc test. The testing level (α) was 0.05. Statistical analyses were performed on all available data, using GraphPad, Prism 7.

## 3. Results

### 3.1. Antiviral Activity of Single Drugs against Two Influenza A Strains

As shown in Table 1, effective dose responses for influenza A(H1N1)pdm09 and A(H3N2) replication were determined for each drug. The EC50 values of the wild-type A(H1N1)pdm09 strain were 3.87 ± 0.36 and 4.05 ± 0.88 µM for ribavirin and favipiravir, respectively. Oseltamivir (0.10 ± 0.05 µM) and zanamivir (0.13 ± 0.07 µM) had the same potency. Peramivir and BXA had the strongest inhibitory activity against A/California/7/2009 virus with EC50 values of 15.00 ± 5.77 and 0.48 ± 0.22 nM, respectively. The EC50 value of oseltamivir and favipiravir were 0.42 ± 0.29 and 10.32 ± 1.89 µM against the A(H3N2) virus, respectively. Similar activities were observed for ribavirin and zanamivir with EC50 values of 2.22 ± 1.55 and 2.48 ± 0.96 µM, respectively. Peramivir (48.43 ± 21.83 nM) and BXA (19.55 ± 5.66 nM) also had the most potent activity against the A(H3N2) strain.

### 3.2. In Vitro Two-Drug Combination Activity against Two Influenza A Strains

Combination drug experiments against the A(H1N1) strain (Table 2A) showed that BXA exhibits synergistic effect when combined with each NAI. The combination index weights (CIwt) were 0.40, 0.48 and 0.48 for zanamivir, oseltamivir and peramivir, respectively. Additionally, BXA and favipiravir had synergistic effects against influenza A(H1N1) with a CIwt of 0.54. However, BXA in combination with ribavirin induced an antagonist effect with a CIwt of 1.91.

In the A(H3N2) background (Table 2B), BXA also demonstrated synergistic effect when combined with each NAI. The CIwt were 0.47, 0.49 and 0.42 for zanamivir, oseltamivir and peramivir, respectively. Furthermore, the combination of BXA and favipiravir also had a highly synergistic activity with a CIwt value of 0.16. Again, the combination of BXA and ribavirin led to an antagonist effect with a CIwt of 1.23.

### 3.3. Two-Drug Combination Activity in Influenza A-Infected HAE

Based on the results obtained in ST6-GalI-MDCK cells, we further evaluated the efficacy of the selected two-drug combination ratios determined in Table 2A in the MucilAirTM reconstituted HAE infection model [32]. Note that different MOIs, 0.1 and 0.01, for A/California/7/2009 (H1N1) and A/Texas/50/2012 (H3N2) viruses, respectively, were based on previous characterization experiments and were used to reflect a comparable replication kinetics curve for both subtypes. As shown in Figure 1, A(H1N1) apical viral titers peaked at 48 hpi in the untreated group, reaching mean values of 4.4 (±3.6) × 108 TCID50/mL. As expected, treatment with 10 nM BXA was strongly effective by inducing a 3 log10 reduction in mean peak viral titers, reaching 5.6 (±4.3) × 105 TCID50/mL. Treatment with 2.7 µM zanamivir (ratio 1:270), 2.1 µM oseltamivir (ratio 1:213) or 310 nM peramivir (ratio 1:31) induced 1.4 log10 (Figure 1A), 3 log10 (Figure 1B) or 2 log10 (Figure 1C) reductions in mean peak viral titers, respectively. Conversely, treatment with 83 µM favipiravir (ratio 1:8333) induced a 24 h delay on peak viral titers, which were comparable to those of the untreated group (Figure 1D). Most importantly, the antiviral effects of the four two-drug combinations tested were systematically higher than those of single-drug treatments, notably in the case of BXA + NAI (zanamivir or oseltamivir), which showed at least a 1 log10 reduction in mean viral titers when compared to the most effective of the single-drug treatments at all time points. The antiviral effect of both single-drug and combination treatments was further supported by TEER values, which remained stable all throughout the experiment, contrary to what was observed for the mock-treated controls. Of note, the extreme disparity of the EC50 ratio obtained for the combination of BXA and ribavirin (1:10,000) hampered its evaluation under experimental conditions in which the ribavirin concentration was sufficiently low not to induce a very significant antiviral effect. 

In the case of A(H3N2), apical viral titers peaked at 72 hpi, reaching mean values of 7.5 (±6.4) × 109 TCID50/mL, in the untreated group (Figure 2). Interestingly, treatment with 10 nM BXA almost completely inhibited viral replication (), for which this dosage was too high to be compatible with the evaluation of drug combination in the A(H3N2) HAE model. We therefore used 5 nM BXA as the reference treatment, which induced a 1.5 log10 reduction in mean peak viral titers, reaching 4.9 (±1.78) × 108 TCID50/mL at 72 hpi. Treatment with zanamivir 625 nM (ratio 1:125) and peramivir 12.35 nM (ratio 1:2.47) induced significant reductions in viral production, as evidenced by 1.7 log10 (Figure 2A) and 1.3 log10 (Figure 2C) lower mean peak viral titers compared to those of the mock-treated controls, respectively. On the other side, single-drug treatment with oseltamivir 105 nM (1:21), favipiravir 2.63 µM (ratio 1:526) and ribavirin 555 nM (ratio 1:111) (Figure 2B,D,E) demonstrated relatively mild antiviral effects, inducing less than 0.5 log10 reductions in mean peak viral titers. Similar to what we observed for A(H1N1)pdm09, the antiviral effects of the different two-drug combinations tested against A(H3N2) were higher than those of single-drug treatments. Of note, the observed differences were statistically significant. The best synergy was obtained with the BXA + oseltamivir combo, which showed a 3 log10 reduction at 48 hpi and a ≥2 log10 reduction at 72 hpi in mean viral titers compared to the mock-treated but also the single-drug treatment conditions. Moreover, BXA + zanamivir and BXA + peramivir improved single-drug treatment at 72 hpi by almost 2 log10 and 1 log10, respectively. TEER measurements further supported these observations, as they remained stable for BXA + oseltamivir, BXA + zanamivir and BXA + peramivir throughout the experiments, which was not the case for single-drug treatments other than BXA. Interestingly, despite the almost negligible antiviral effect of favipiravir and ribavirin single-drug treatments, both BXA + favipiravir and BXA + ribavirin showed 2.8 log10 and 2.2 log10 reductions in viral titers at 48 hpi, respectively. Nevertheless, such an effect was not observed at 72 hpi, which is in line with the reduction in TEER values observed at this time point.

## 4. Discussion

Antivirals play a crucial role in the treatment and control of influenza epidemics and pandemics [33]. The M2 ion-channel blockers (adamantanes including amantadine and rimantadine) [34], NAIs (oseltamivir, zanamivir, peramivir or laninamivir) [35] and the polymerase complex inhibitors (BXM or favipiravir) have demonstrated clinical benefits, but the emergence of viral resistance may limit the clinical efficacy of monotherapies [36,37]. Antiviral combinations have been suggested to prevent or delay the emergence of resistance by targeting various steps in the virus replicative process [38]. Multiple therapies may allow for dosage reduction [33], in addition to improving therapeutic effect [38]. Drug combinations already exist to treat cancer [39] and infectious diseases caused by HIV and hepatitis C [40,41,42,43]. In a clinical study, the triple combination of oseltamivir, amantadine and ribavirin has been reported to decrease influenza viral shedding compared to monotherapy [24]. 

In this study, we aimed at investigating the potential benefits of combining BXA to most approved antivirals against seasonal influenza A(H1N1) and A(H3N2) strains using both a classic cell line and a physiologically and predictively relevant HAE model.

Firstly, we observed that BXA had the lowest EC50 (Table 1) against A(H1N1) and A(H3N2) viruses. These effective concentrations of BXA were followed by those of NAIs and other polymerase inhibitors. Of note, EC50 values were higher for A(H3N2) than for A(H1N1). Depending on the antiviral, effective concentrations could differ by 2.5- to 39-fold between strains. Additionally, BXA susceptibilities of influenza viruses have been shown to vary depending on viral strain or subtype [20,44]. Therefore, determinations of EC50 value for each drug were the first step in the systematic evaluation of drug combinations by using an equipotent ratio between BXA and other antivirals.

Secondly, our results show that combinations of BXA with NAIs (oseltamivir, peramivir or zanamivir) have higher potency compared to individual drugs, against A(H1N1)pdm09 and A(H3N2) viruses. Such synergistic effects were confirmed in the HAE model, for which the physiological relevance and predictive value have been previously described by our group and others [33]. These findings are in agreement with those reported by Fukao et al., where in vitro combinations of BXA and approved NAIs also resulted in a synergistic effect against the A/PR/8/34 (H1N1) virus [26]. In that study, the authors further showed that the combination of suboptimal doses of BXM and oseltamivir enhanced therapeutic effects in mice, including reduced lung viral titers. 

Combinations of other polymerase inhibitors with NAIs, as well as combinations of two polymerase inhibitors, have also been studied. Ormond and colleagues demonstrated that the combination of oseltamivir with favipiravir negatively impacted viral replication after several passages [45]. These results are in accordance with different animal studies where peramivir–favipiravir [46] or oseltamivir–favipiravir [13] combinations provided various benefits, including improving survival, controlling drug-resistant A(H1N1) strains or delaying mortality in immunocompromised animals. 

In our study, we also assessed two polymerase inhibitors: favipiravir and ribavirin (a transcriptase inhibitor of Polymerase Basic 1—PB1) [47,48] in combination with BXA. We obtained synergistic effects for the BXA-favipiravir combination against the two influenza A strains in cell culture, which were further observed in HAE mainly in the case of A(H3N2). In contrast, the BXA-ribavirin combination showed mixed results compared to single-drug treatment, depending on the infection model and viral subtype used. In fact, considering that BXA and ribavirin compounds inhibit two polymerase subunits (PA and PB1, respectively), we suggest that these two agents targeting similar but non-identical functions can interfere or compete in their mechanism of action, resulting in antagonism. Indeed, the PA and PB1 subunits, which interact extensively [49], could be in close proximity during the transcription/replication process, which may limit their accessibility to the two inhibitors simultaneously. In a modeling drug action experiment, Yin et al. [50] demonstrated that, when used in combination, drugs interact in many unexpected ways and show a plethora of different outcomes, leading to drug synergy or antagonism. Biological functions involve many molecules that interact in a network manner. Thus, the final effects of drug combinations also depend on the interactions of their targets in a network manner [50], especially in physiological models, such as HAE, which is constituted by primary differentiated human cells mimicking in vivo epithelia. Indeed, a household study showed that the combination of oseltamivir and zanamivir, administrated within 24 h of onset of symptoms, was more effective in reducing the transmission of influenza compared to monotherapy [51], whereas we reported that the same combination was not more protective than zanamivir monotherapy against A(H1N1)pdm09 and A(H3N2) infections in mice [52]. This difference could be explained by the severity of influenza infections (uncomplicated versus severe infections). Moreover, the optimal dose of each component greatly affects drug interactions [53]. Our combination experiments were based on a fixed ratio calculation. We hypothesize that this method could not apply to the combination of BXA and ribavirin and that a different ratio may lead to another type of drug response. Indeed, different dose combinations may lead to different effects [54]. Interestingly, in some cases, drugs with antagonist effects could also be beneficial against resistant viral strains [50,55]. 

Additional in vitro studies using resistant strains (I38T or H275Y variants) should be done to better understand drug activities and to evaluate the impact of combination therapy on mutant viral replication. Additionally, further animal studies should be conducted to evaluate the effectiveness of our synergistic combinations in a complete system.

In summary, our results suggest that BXM stands as a potential good candidate for combination therapy with several existing drugs to improve antiviral activity and eventually delay drug resistance of influenza type A viruses. Combination therapies with BXM should be further considered for animal studies and clinical investigations.

## Figures and Tables

**Figure 1 viruses-12-01139-f001:**
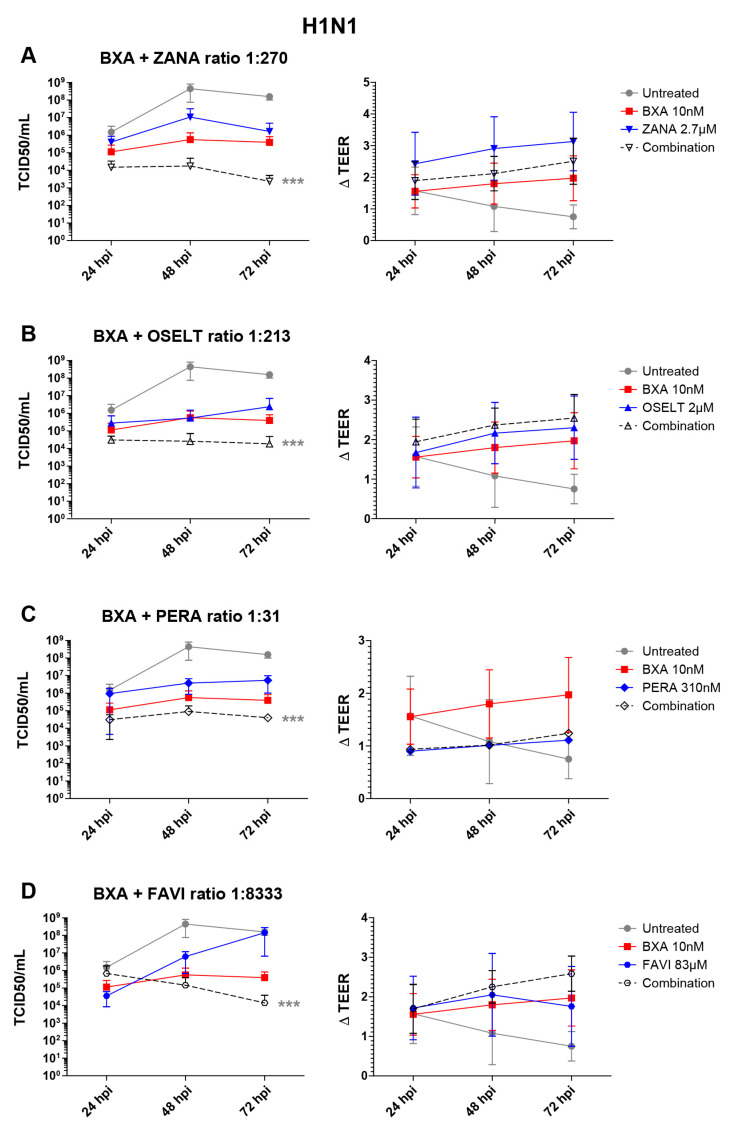
Two-drug combination activity in influenza A(H1N1)-infected reconstituted human airway epithelia (HAE). Apical viral production (TCID50/mL ± standard deviation, SD) and transepithelial electrical resistance (ΔTEER (transepithelial electrical resistance) vs. *t* = 0 ± SD) in MucilAirTM HAE infected on the apical pole with influenza A/California/7/2009 (H1N1) virus at a multiplicity of infection (MOI) of 0.1 and treated with the indicated antiviral combinations and their corresponding single drug controls by the basolateral pole: (**A**) baloxavir acid + zanamivir; (**B**) baloxavir acid + oseltamivir; (**C**) baloxavir acid + peramivir; (**D**) baloxavir acid + favipiravir. *** *p* < 0.001 compared to the infected untreated group using mixed model two-way analysis of variance (ANOVA) with Bonferroni post hoc test. Data are representative of at least four independent experiments.

**Figure 2 viruses-12-01139-f002:**
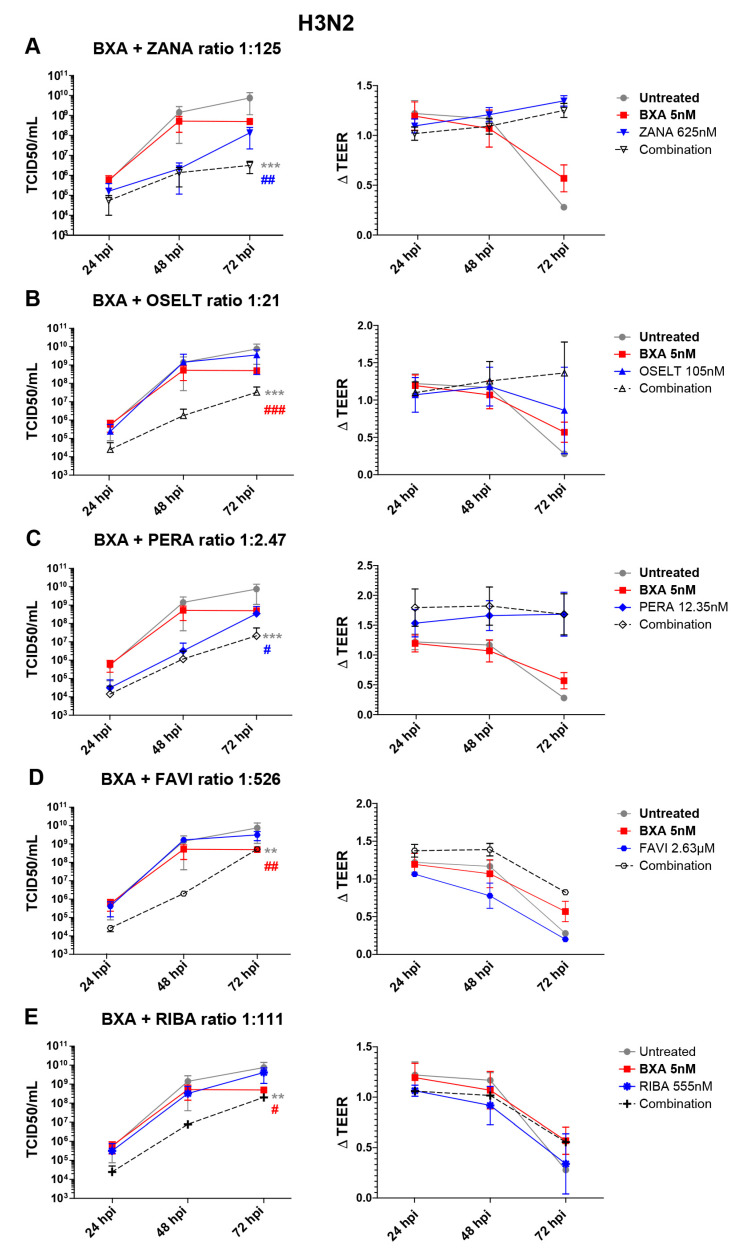
Two-drug combination activity in influenza A(H3N2)-infected reconstituted human airway epithelia (HAE). Apical viral production (TCID50/mL ± SD) and transepithelial electrical resistance (ΔTEER vs. *t* = 0 ± SD) in MucilAirTM HAE infected on the apical pole with influenza A/Texas/50/2012 (H3N2) virus at a MOI of 0.01 and treated by the basolateral pole with the indicated antiviral combinations and their corresponding single drug controls: (**A**) baloxavir acid + zanamivir; (**B**) baloxavir acid + oseltamivir; (**C**) baloxavir acid + peramivir; (**D**) baloxavir acid + favipiravir; (**E**) baloxavir acid + ribavirine. ** *p* < 0.01 and *** *p* < 0.001 and ^#^
*p* < 0.05, ^##^
*p* < 0.01 and ^###^
*p* < 0.001 compared to the infected untreated or the most performant single-treated groups (BXA (# in red) and ZANA, or PERA (# in blue)), respectively, using mixed model two-way analysis of variance (ANOVA) with Bonferroni post hoc test. Data are representative of at least three independent experiments.

**Table 1 viruses-12-01139-t001:** Antiviral activity of individual drugs against two influenza A strains.

	Effective Concentration That Inhibits Virus Effect by 50% (nM)	
Antiviral	A/California/7/2009 (H1N1)pdm09	A/Switzerland/9715293/2013 (H3N2)
Baloxavir acid	0.48 ± 0.22	19.55 ± 5.66
Ribavirin	3872.65 ± 356.54	2222.10 ± 1556.82
Oseltamivir	101.67 ± 54.19	420.00 ± 287.17
Zanamivir	132.00 ± 74.63	2475.00 ± 962.27
Favipiravir	4050.00 ± 880.83	10,323.33 ± 1889.19
Peramivir	15.00 ± 5.77	48.43 ± 21.83

Data are means EC50 (50% effective concentration) ± standard deviation (SD) from at least three independent experiments.

**Table 2 viruses-12-01139-t002:** In vitro two-drug combination activity against (**A**) A(H1N1) and (**B**) A(H3N2) strains.

**(A)**	**A/California/7/2009 (H1N1)pdm09**						
		**CI values extrapolated at % of virus inhibition ^b^**					
**Antiviral combination**	**EC50 ratio ^a^**	**50**	**75**	**90**	**95**	**CIwt ^c^**	**Effect ^d^**
BXA + ZANA	1:270.27	1.00 ± 0.52	0.49 ± 0.28	0.32 ± 0.16	0.26 ± 0.11	0.40	synergism
BXA + OSELT	1:212.7	0.80 ± 0.56	0.53 ± 0.37	0.43 ± 0.22	0.40 ± 0.17	0.48	synergism
BXA + PERA	1:31.06	0.86 ± 0.27	0.53 ± 0.35	0.43 ± 0.41	0.40 ± 0.43	0.48	synergism
BXA + RIBA	1:10,000	1.56 ± 0.39	1.63 ± 0.49	1.88 ± 0.89	2.16 ± 1.27	1.91	antagonism
BXA + FAVI	1:8333	0.34 ± 0.06	0.36 ± 0.23	0.51 ± 0.48	0.69 ± 0.79	0.54	synergism
**(B)**	**A/Switzerland/9715293/2013 (H3N2)**						
		**CI values extrapolated at % of virus inhibition ^b^**					
**Antiviral combination**	**EC50 ratio ^a^**	**50**	**75**	**90**	**95**	**CIwt ^c^**	**Effect ^d^**
BXA + ZANA	1:125	0.65 ± 0.42	0.36 ± 0.32	0.41± 0.60	0.52 ± 0.84	0.47	synergism
BXA + OSELT	1:21.28	0.67 ± 0.48	0.53 ± 0.23	0.46 ± 0.01	0.45 ± 0.13	0.49	synergism
BXA + PERA	1:2.46	0.28 ± 0.07	0.31 ± 0.1	0.40 ± 0.27	0.51 ± 0.46	0.42	synergism
BXA + RIBA	1:111	4.11 ± 2.23	1.52 ± 0.50	0.84 ± 0.82	0.65 ± 0.78	1.23	moderate antagonism
BXA + FAVI	1:125	0.21 ± 0.07	0.16 ± 0.04	0.15 ± 0.04	0.16 ± 0.07	0.16	synergism

^a^ Ratio of EC50 values (effective concentrations that inhibit cytopathic effects of each virus by 50%) of each drug alone (that is EC50 drug_1_/EC_50_ drug_2_). ^b^ Combination index (CI) values extrapolated at indicated % of virus inhibition by use of the Compusyn software. CI values represent the means ± standard deviation of two to three independent experiments done in duplicate. ^c^ CIwt weighted average CI values were calculated as (CI_50_+ 2 × CI_75_ + 3 × CI_90_ + 4 × CI_95_)/10. ^d^ Drug combination effects were defined as CIwt < 0.7, synergism; CIwt > 0.7 and <0.9, moderate synergism; CIwt > 0.9 and <1.2, additivity; CIwt > 1.2 and <1.45, moderate antagonism and CIwt > 1.45, antagonism. BXA, baloxavir acid; ZANA, zanamivir; OSELT, oseltamivir carboxylate; PERA, peramivir; RIBA, ribavirin; FAVI, favipiravir.

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
