# Peer review of "In Vitro Combinations of Baloxavir Acid and Other Inhibitors against Seasonal Influenza A Viruses"

_viruses, 2020, doi:10.3390/v12101139_

Round 1
Reviewer 1 Report
My previous concerns have been fully eliminated, no further questions.
Author Response
We would like to thank the reviewer for his positive comments.
Reviewer 2 Report
- There are differences in the experimental settings in Figures 1 and 2 (i.e., differences in a multiplicity of infection (MOI) and concentrations of neuraminidase and polymerase inhibitors). Why? The authors should explain this in the Results section.
- Page 8, lines 203-205. The authors claim that “In the case of A(H3N2), the observed viral replication kinetics was slightly slower than that of A(H1N1)”. However, as mentioned above, A/California/7/2009 (H1N1pdm09) and A/Texas/50/2012 (H3N2) were inoculated into human airway epithelia at an MOI of 0.1 and 0.01, respectively (Figures 1 and 2). To compare the growth kinetics between the two influenza A virus strains, the inoculated viruses should be equal in MOI.
- Figures 1 and 2. The authors performed statistical analyses of virus titers in human airway epithelia. At what time point after infection was a significant difference observed? They should show an asterisk or other symbol to indicate statistical significance at each time point (i.e., at 24, 48, and 72 post-infection).
Author Response
- There are differences in the experimental settings in Figures 1 and 2 (i.e., differences in a multiplicity of infection (MOI) and concentrations of neuraminidase and polymerase inhibitors). Why? The authors should explain this in the Results section.
It is usual that different viral subtypes behave differently in terms of viral kinetics, with H3N2 strains generally showing higher and/or faster replication rates than H1N1pdm09 variants at the same MOI. The MOIs used in this study (page 5; lines 172-174) stem from several previous characterization experiments of both viral strains in the HAE model, in which we found that the A/Texas/50/2012 (H3N2) strain had a replication kinetics curve comparable to that of the A/California/7/2009 (H1N1pdm09) using a tenfold lower MOI.
An analogous rationale applies for the choice of the BXA concentration: H3N2 and H1N1 viruses are not equally susceptible to BXA, which is reflected by the differential effect of the 10 nM BXA dose. In order to evaluate the potential benefit of drug combinations in HAE, we had to choose a BXA concentration that did not completely abrogate viral replication. We therefore halved the BXA dose to 5 nM for experiments with the H3N2 virus, but always keeping constant the drug ratios obtained in cell-based experiment, not to derive from the original hypothesis.
- Page 8, lines 203-205. The authors claim that “In the case of A(H3N2), the observed viral replication kinetics was slightly slower than that of A(H1N1)”. However, as mentioned above, A/California/7/2009 (H1N1pdm09) and A/Texas/50/2012 (H3N2) were inoculated into human airway epithelia at an MOI of 0.1 and 0.01, respectively (Figures 1 and 2). To compare the growth kinetics between the two influenza A virus strains, the inoculated viruses should be equal in MOI.
The sentence was not aimed at specifically comparing the two viruses but at describing each replication curve in the experimental conditions of the study. Nevertheless, we omitted the sentence as requested (page 7, lines 201-202).
- Figures 1 and 2. The authors performed statistical analyses of virus titers in human airway epithelia. At what time point after infection was a significant difference observed? They should show an asterisk or other symbol to indicate statistical significance at each time point (i.e., at 24, 48, and 72 post-infection).
We did not perform a specific statistical analysis for each time point. As requested by reviewer 3, we used a two-way ANOVA test that compares each curve as a whole. We believe this approach provides higher robustness to the analysis, for which we prefer to keep it that way.
Round 2
Reviewer 2 Report
I have no comments on the revised version.
This manuscript is a resubmission of an earlier submission. The following is a list of the peer review reports and author responses from that submission.
Round 1
Reviewer 1 Report
The work of Liva Checkmahomed et al. shows that the use of combined therapy of BXA and other commonly used antiviral drugs against influenza A viruses can increase their potency by synergic effect, at least in in-vitro experiments. Also their results showed that BXA had antagonist effect if it has been used in combination with RdRp related inhibitors, such as ribavirin. The manuscript is, in general, well written and clearly structured. However, some issues must be revised:
Materials and methods:
- The authors did not describe why they used these specific dilutions of the antiviral drugs for testing in cell culture. It is necessary to include if these dilutions were given by the manufacturer where they purchased the drugs, or if they followed specific criteria based on previous publications.
- The HAE approximation is very interesting. However, the authors only performed this experiment against the A/California/07/2009 strain and not against A/Switzerland/9715293/13. It will be very interesting to perform also against this last strain, and if the authors have this data, they must include in this manuscript. If this data is not performed because any reason, the authors must include this reason in the manuscript.
Discussion:
- The first paragraph of the discussion (lines 187-202) is reiterative and must be included in the introduction. In this paragraph the authors did not talk about their own results, they only introduce some information about antivirals that can be included in the introduction.
- Despite the great importance that have the experiments performed in the HAE model, that is a great approximation to what occurs in the airway epithelial cells, the authors only mention this experiment in one phrase in the discussion (line 208). No other mention to this experiment throughout the discussion was found. Also, the authors did not mention why they did not use this assay against A(H3N2) virus. The authors must write extended discussion focusing in this point and their relevance for the results.
- The A(H3N2) is the great “disadvantaged” in this manuscript. The results showed that the effective concentration for inhibiting the virus effect in 50% using individual antiviral drugs was several times greater for A(H3N2) than for A(H1N1)pdm09. This must be mentioned in the discussion, and the authors must try to explain why the antiviral drugs seems to be less effective against A(H3N2) than the 2009 strain. This point has great implications in drug delivery and dosage, and also for assessing the ability for combining different antiviral drugs in the same therapy.
Reviewer 2 Report
Checkmahomed et al., “In vitro combinations of baloxavir acid and other inhibitors against seasonal influenza A viruses”
In this manuscript, the authors assessed efficacies of baloxavir acid, the active form of baloxavir marboxil, combined with other approved antiviral drugs (i.e., neuraminidase inhibitors, favipiravir, and ribavirin) against influenza A virus infection in vitro. They show that the combination of baloxavir acid with neuraminidase inhibitors or favipiravir inhibits more efficiently the replication of influenza A/H1N1 2009 pandemic and A/H3N2 viruses compared to single drug treatment. However, their data are insufficient to support the claim. At the very least the authors should evaluate the combination effect of baloxavir acid with NA inhibitors or favipiravir in vivo. Data on the effect of baloxavir acid in combination with favipiravir or ribavirin are new; however, two-drug synergistic inhibition of influenza A virus replication by baloxavir acid and neuraminidase inhibitors has previously been reported elsewhere (ref. 26). Therefore, the amount of novel information present in this study is limited.
Specific comments:
- Fig. 1: Since only A/California/7/2009 (H1N1)pdm09 was tested, it is not clear whether the combination of baloxavir acid with NA inhibitors or favipiravir is uniformly synergistic against other influenza A virus subtypes. The authors should also examine influenza A/H3N2 viruses. In addition, they need to show statistical analyses in terms of virus titer and transepithelial electrical resistance.
- The authors could give more explanation as to why they chose two influenza A virus strains (A/California/7/2009 (H1N1)pdm09 and A/Switzerland/9715293/2013 (H3N2)) for testing antiviral activity of drugs.
- Regarding ribavirin, no wide‐scale authorizations have been made for its use against influenza. Why did the authors select this drug?
- Line 76: A/California/7/2009 (H1N1) should be A/California/7/2009 (H1N1)pdm09.
- Line 76: A/Switzerland/9715293/13 (H3N2) should be A/Switzerland/9715293/2013 (H3N2).
Reviewer 3 Report
The reviewer appreciated reading this study but would strongly recommend the authors to address the issues the reviewer mentioned as follows to improve the impact of the study.
- Page 3 Lines 120-123. Were any TEER changes observed after treatment of each molecule alone or in combination without A/California/7/2009 inoculation? In Figure 1, each molecule alone or in combination gave different effect to TEER of MucilAir in the presence of A/California/7/2009 and each effect was not parallel to each molecule-mediated reduction in virus titers. Nothing about TEER was described in results section, thus the reviewer has difficulty in understanding the effect of molecules to TEER and the relationship between virus growth and epithelium integrity. Please explain about them in result section and/or in discussion section.
- In section 3.3 and Figure 1, the authors presented the efficacy of molecules using MucilAir. The reviewer recommends the authors to perform a statistical analysis to evaluate the significance of molecules-mediated reduction in virus titers. Two-way ANOVA might be applied.
- Zanamivir is an inhalant. Did treatment with Zanamivir through the apical poles of MucilAir induce reductions in viral titers? And were any differences observed between treatment through the apical poles and basolateral poles?
- Page 8 Lines 249-266. Please describe Author Contributions, Funding, and Conflicts of Interest.